# The Role of Extracellular Vesicles in Ischemic Stroke Severity

**DOI:** 10.3390/biology11101489

**Published:** 2022-10-11

**Authors:** Angelica Carandina, Chiara Favero, Roberto Maria Sacco, Mirjam Hoxha, Giuseppe Torgano, Nicola Montano, Valentina Bollati, Eleonora Tobaldini

**Affiliations:** 1Department of Internal Medicine, Fondazione IRCCS Ca’ Granda, Ospedale Maggiore Policlinico, 20122 Milan, Italy; 2Department of Clinical Sciences and Community Health, University of Milan, 20122 Milan, Italy; 3Department of Anesthesia, Critical Care and Emergency, Fondazione IRCSS Ca’ Granda, Ospedale Maggiore Policlinico, 20122 Milan, Italy; 4Occupational Health Unit, Fondazione IRCCS Ca’ Granda, Ospedale Maggiore Policlinico, 20122 Milan, Italy

**Keywords:** ischemic stroke, extracellular vesicles, platelet-derived extracellular vesicles, T-cell-derived extracellular vesicles

## Abstract

**Simple Summary:**

Ischemic stroke represents one of the leading causes of death and disability worldwide. The identification of new prognostic factors and biomarkers for patients’ risk stratification could reduce the burden of disease. In this perspective, given the possibility of non-invasively collecting the extracellular vesicles and characterizing them on the basis of parental surface markers, we verified whether extracellular vesicles could represent an interesting prognostic biomarker in ischemic stroke. We found that specific extracellular vesicle subtypes are associated with stroke severity and both short- and long-term outcomes.

**Abstract:**

The possibility of characterizing the extracellular vesicles (EVs) based on parental cell surface markers and their content makes them a new attractive prognostic biomarker. Thus, our study aims to verify the role of EVs as relevant prognostic factors for acute and mid-term outcomes in ischemic stroke. Forty-seven patients with acute ischemic stroke were evaluated at admission (T0), immediately after recanalization treatment or after 2 h in non-treated patients (T1) and after one week (Tw) using the National Institutes of Health Stroke Scale (NIHSS), and after 3 months using the Modified Rankin Scale (mRS). Total count and characterization of EVs were assessed by Nanosight analysis and flow cytometry. The relationships between stroke outcomes and EV count were assessed through multivariable negative binomial regression models. We found that the amount of platelet-derived EVs at admission was positively associated with the severity of ischemic stroke at the onset as well as with the severity of mid-term outcome. Moreover, our study revealed that T-cell-derived EVs at admission were positively related to both early and mid-term ischemic stroke outcomes. Finally, T-cell-derived EVs at T1 were positively related to mid-term ischemic stroke outcome. The present study suggests that specific EV subtypes are associated with stroke severity and both short- and long-term outcomes. EVs could represent a valid tool to improve risk stratification in patients with ischemic stroke and post-recanalization treatment monitoring.

## 1. Introduction

Extracellular vesicles (EVs) are membrane-surrounded particles released from living cells into the extracellular space, whose size ranges from 30 nm to 1 µm. EVs play an important role in several physiological and pathological processes, contributing to intercellular communication and acting as a transporter of bioactive molecules (such as proteins, lipids, mRNA, and miRNA) from the tissue of origin to target cells [1,2]. The possibility of collecting EVs from non-invasive biofluids (e.g., plasma) and characterizing the EVs based on parental surface markers and their content makes them a new attractive prognostic biomarker and a useful diagnostic tool. Recent research has shown a possible link between plasmatic EVs and both cardiovascular diseases and cardiovascular risk factors (e.g., hypertension, obesity) [3,4].

Cardiovascular diseases are the leading cause of death globally, and among these, stroke is the second major killer [5]. In Europe, about 1.1 million inhabitants suffered a stroke each year, and ischemic stroke represents the main subtype with approximately 80% of cases [6]. The one-month fatality rate of ischemic stroke has been estimated at around 15% in high-income countries [7], and the percentage of persistent residual disability at three months after the event is higher than 50% in the survivors [8]. Even though pharmacological and mechanical recanalization treatments are the most effective therapeutic approaches for patients with acute ischemic stroke (AIS) and many efforts have been made for the development of prognostic factors and the identification of biomarkers for patients’ risk stratification, the burden of ischemic stroke in terms of mortality-to-incidence ratio and disability-adjusted life-years (DALYs) remains challenging [9]. From this perspective, our study aims to verify the role of EVs as a relevant prognostic factor for acute and mid-term outcomes in ischemic stroke.

## 2. Materials and Methods

### 2.1. Study Population

From September 2016 to March 2018, 47 patients with a diagnosis of AIS were admitted to in the Emergency Department of Fondazione IRCCS Ca’ Granda, Ospedale Maggiore Policlinico (Milan, Italy). We included patients presenting new neurological symptoms within 4.5 h and for at least 30 min. Subjects over 18 years old and with stable spontaneous sinus rhythm on the ECG at presentation were eligible for the study. Exclusion criteria were: primary intracerebral hemorrhage, pre-existing neurological conditions, epilepsy at the onset of stroke symptoms, severe organ failure, active oncological conditions, mechanical ventilation, or consent refusal. Demographics, anthropometrics, clinical data, and standard laboratory evaluations were collected at admission. All subjects signed the informed consent. The study was conducted in accordance with the Declaration of Helsinki, and the protocol was approved by the local Ethics Committee “Comitato Etico—Milano Area 2”, Milan, Italy (approval code 1443/2016).

### 2.2. Experimental Protocol

All the enrolled patients underwent the diagnostic and therapeutic process as required by the International and Internal Guidelines for the treatment of acute ischemic stroke [10]. The National Institutes of Health Stroke Scale (NIHSS) was used to assess stroke severity. NIHSS is based on the evaluation of 11 neurological items, it ranges from 0 to 42 (lower scores indicate less severe neurological impairment). NIHSS was evaluated at admission (T0) and after one week (Tw) [11]. A NIHSS score ≥ 14 refers to severe conditions [11]. To collect a mid-term outcome and to evaluate the residual functional disability, the patients underwent a follow-up assessment after 3 months through the Modified Rankin Scale (mRS), which ranges from 0 (patient without symptoms) to 6 (death). Dichotomous cut-offs of 0–2 versus 3–5 were used to categorize patients in two categories of disability severity [12].

According to the guidelines, intravenous (IV) thrombolytic therapy (r-tPA at 0.9 mg/kg; 90 mg as maximum total dose; 10% as IV bolus and the remainder infused over 60 min) was planned for enrolled patients without previous or current cerebral hemorrhage or bleeding disorder or coagulopathy or prolonged activated partial thromboplastin time (aPTT), who did not receive heparin in the previous 48 h. Other relative contraindications were also considered [13,14]. Mechanical thrombectomy was planned for AIS patients with large-vessel occlusion who received intravenous r-tPA at admission, if allowed, with symptoms onset within 6 h or in case of contraindications to IV r-tPA [13,14,15].

Blood samples were collected in ethylenediaminetetraacetic acid (EDTA) tubes (7 mL) at admission (T0), immediately after any reperfusion therapy (thrombolytic drug administration/thrombectomy) or 2 h after the first blood sample (T1), and after one week (Tw). The study protocol is illustrated in Figure 1. 

### 2.3. EV Isolation

Transport to the EPIGET Lab (University of Milan) and processing of blood samples was performed within two hours of collection. EDTA-blood was centrifuged at 1200× *g* for 15 min at room temperature to obtain platelet-free plasma. Plasma was further centrifuged at 1000, 2000, and 3000× *g* for 15 min at 4 °C, and the resulting pellets of cell debris were discarded. EV preparation and analyses were done in compliance with the MISEV 2018 guidelines (Detailed in Appendix A). To prepare EV pellets for Nanosight analysis and flow cytometry, 1.5 mL of fresh plasma was transferred to an ultracentrifuge tube (Quick-Seal, Round-Top, polypropylene, 13.5 mL; Beckman Coulter, Inc., Indianapolis, IN, USA), which was then filled up with phosphate-buffered saline (PBS) previously passed 3× through a sterile membrane filter unit (Stericup-VP, 0.10 µm, polyethersulfone; EMD Millipore, Billerica, MA, USA) in order to reduce background interferers as much as possible. Plasma was then spun in a benchtop ultracentrifuge (Optima MAX-XP; Beckman Coulter, Inc.) at 110,000× *g* for 75 min at 4 °C, to obtain an EV-rich pellet. The pellet was resuspended with 500 µL of triple membrane 0.1 µm filtered PBS for further analysis.

### 2.4. Nanosight Analysis

The number and dimension of EVs were assessed by nanoparticle tracking analysis (NTA). This technique measures the Brownian motion of vesicles suspended in a fluid and displays them in real time through a charge-coupled device camera with high sensitivity. The Nanosight LM10-HS system (NanoSight Ltd., Amesbury, U.K.), was used to visualize the EVs by laser light scattering. A total of 5 30 sec recordings were performed for each sample. The size distribution profiles and EV concentration measurements were provided through NTA software. EVs were expressed in scientific notation N × 10^6^ for 1 mL of plasma.

### 2.5. Flow Cytometry

EV subtypes were characterized with the MACSQuant Analyzer 10 flow cytometer (Miltenyi Biotec, Calderara di Reno, BO, Italy) according to a customer-provided protocol (https://bit.ly/3BuB6ze, accessed on 7 September 2022). The Fluoresbrite Carboxylate Size Range Kit I (0.2 µm, 0.5 µm, 0.75 µm, and 1 µm) was used to set the calibration gate on the MACSQuant Analyzer system. To evaluate EV integrity, 60 µL sample aliquots were stained with 0.02 µM 5(6)-carboxyfluorescein diacetate N-succinimidyl ester (CFSE) at 37 °C for 20 min in the dark (Pospichalova et al., 2015). CFSE is a vital dye non-fluorescent molecule able to enter into EV, where intracellular esterases remove the acetate group and convert the molecule into the fluorescent ester form. To perform the EV subtyping, specific antibodies were added to each aliquot of CFSE-stained sample: CD14-APC (Clone TÜK4) to distinguish EVs derived from macrophages and/or monocytes, CD61-APC (clone: Y2/51) to distinguish EVs derived from platelets, CD105-APC (clone: 43A4E1) to distinguish EVs derived from total endothelial cells, CD25-APC (clone: 4E3) to distinguish EVs derived from T-cells, and CD62e-APC (clone REA280) to distinguish EVs derived from activated endothelial cells. All antibodies were purchased from Miltenyi Biotec. All conjugated antibodies were diluted 1:5 in triple-filtered PBS and then centrifuged at 17,000× *g* for 30 min at 4 °C to eliminate aggregates. All antibodies were used at a final dilution of 1:50 according to the manufacturer’s instructions. A stained PBS (control) sample was used to detect the autofluorescence of the antibody. Quantitative multiparameter analysis of flow cytometry data (expressed as 10^3^ for 1 mL of plasma) was carried out using FlowJo software (Tree Star, Inc., Ashland, OR, USA). This method allowed us to detect EVs ≥ 200 nm.

### 2.6. Statistical Analysis

Descriptive statistics were performed on all variables. Continuous data were expressed as the mean ± standard deviation (SD) or as the median and interquartile range (Q1–Q3), as appropriate. Categorical data were presented as frequencies and percentages.

We applied multivariable negative binomial regression models for over-dispersed count observations to evaluate the relationship between NIHSS and mRS scale and extracellular vesicles (EV) count (total, CD61+, CD14+, CD62E+, CD105+, and CD25+) measured at different times (T0, T1, and TW). We tested the presence of over-dispersion basing upon the Lagrange Multiplier (LM) test. The regression models were adjusted for age, gender, BMI, smoking habit, glucose level, blood pressure, and AIS severity at the onset. In the models with EVs measured at T1 and TW, we adjusted also for therapy. Estimated effects were described as a percentage of variation associated with an increase of 10^6^ in EV total count or an increase of 10^3^ in EV subtype (1-incidence rate ratio (IRR)) *100, IRR = exp (β).

For each EV size from 30 nm to 700 nm, we estimated geometric means and 95% CI of total count EV measured at Tw in patients with or without therapy, with negative binomial regression models adjusted for age, gender, BMI, smoking status, glucose level, and blood pressure. Due to the high number of comparisons, we used a multiple comparison method based on the Benjamini–Hochberg False Discovery Rate (FDR) to calculate the FDR *p*-value. To display the results of the analyses, we used a series graph for means and 95% CI and vertical bar charts to represent FDR *p*-values and *p*-values. For the two graphs, the size of EVs was reported on the *X*-axis.

All statistical analyses were performed with SAS software (version 9.4; SAS Institute Inc., Cary, NC, USA). A two-sided *p*-value of 0.05 was considered statistically significant.

## 3. Results

A total of 47 patients were included in the study (26 males and 21 females). Demographic and clinical characteristics are described in Table 1. The mean age was 74.1 ± 14.1 years and 25.5% of the patients experienced a previous stroke or transient ischemic attack. We evaluated the occurrence of the main cardiovascular risk factors: hypertension (66%), diabetes (≅15%), previous episodes of atrial fibrillation (≅32%), and heart failure (≅13%). A total of 21.3% of patients were current smokers.

The median value of NIHSS on admission was 8 (IQR 5–14). A total of 13 patients had a NIHSS score ≥ 14 and 34 patients had a NIHSS score < 14 [11]. A total of 30 patients underwent recanalization therapy: 18 received IV rtPA, 4 only thrombectomy, and 6 a combination of IV rtPA and thrombectomy. Cardioembolic stroke occurred in ≅49% of patients, ≅13% of patients had an atherothrombotic stroke, and 4.3% had a lacunar stroke. The median value of NIHSS at Tw was three (IQR 2–7). A total of six patients were characterized by severe Tw NIHSS scores and five patients died within one week. The mean value or mRS was 2.8 at 3 months. A total of 20 patients had no or low residual disability, whereas 10 patients were characterized by a severe degree of disability and 7 patients died within 3 months. Patients with hypertension presented higher NIHSS scores after one week, while no significant relationships emerged between smoking status and stroke outcomes (see Appendix A).

Counts of total EVs quantified by Nanosight analysis and counts of EV subtypes obtained by flow cytometry are shown for T0, T1, and Tw in the Appendix A. The median count of total EVs was 4201 × 10^6^/mL plasma, 3196 × 10^6^/mL plasma, and 4300 × 10^6^/mL plasma at T0, T1, and Tw, respectively. As for cellular origin by flow cytometry, the larger EV fraction originated from platelets (CD61+) and smaller proportions of EVs derived from macrophages/monocytes (CD14+), T-cells (CD25+), stationary endothelial (CD105+) cells, and activated endothelial (CD62E+) cells were identified.

Relationships between Total EV concentrations and hypertension and smoke are reported in Appendix A. Hypertension was associated with a higher amount of endothelial-derived EVs at admission (see Appendix A).

Considering the acute assessment, the statistical analysis did not reveal a significant association between the NIHSS score at the onset of AIS and the total amount of EVs collected at T0. However, the NIHSS score at the onset was positively associated with the EV proportion derived from platelets, with approximately a 0.177% increase in NIHSS score for each 10^3^ unit increase in EV subtype count, as reported in Table 2.

Tw NIHSS score was positively associated with T-cell-derived EVs at T0, with a 3.93% increase in NIHSS score for each 10^3^ unit increase in EV subtype count, respectively (Table 3).

Concerning the mid-term outcome shown in Table 4, the statistical analysis did not highlight statistically significant relationships between the mRS score at 3 months and the total count of EVs from all three hematic samples (T0, T1, and Tw). Among the EV subtypes, the association between platelet-derived EVs at T0 and the mRS score was significant, with a percentage increase in the mRS score equal to 0.347 for each 10^3^ increase in platelet-derived MV count. Furthermore, the T-cell-derived EVs from blood samples at T0 and T1 were positively associated with the mRS score after 3 months, with a 3.94% and a 0.91% increase in the mRS score for each 10^3^ unit increase in EV subtype count, respectively.

Finally, as shown in Figure 2, statistical analysis revealed a significant difference in terms of the total amount of EVs from the blood sample collected on TW between patients who underwent therapy and patients who did not undergo therapy, in particular, the former had a smaller quantity of EVs ranging in size from about 70 to 90 nm and from about 160 to 530 nm.

## 4. Discussion

The main finding of the present study on patients with AIS is that specific EV subtypes are associated with stroke severity at different timepoints. In particular: (i) the amount of platelet-derived EVs from the blood sample at admission is positively associated with both the severity of NIHSS score at onset and the severity of the mid-term outcome (mRS score after three months); (ii) the amount of T-cell-derived EVs from the blood sample at admission are positively related to the early and mid-term AIS outcome (NIHSS score after one week and mRS score after 3 months, respectively); (iii) the amount of T-cell-derived EVs from the blood sample collected after therapy/2 h after the event is positively related to the mid-term AIS outcome (mRS score after three months); and (iv) patients who did not undergo therapy showed a higher total amount of EVs with specific size (70 to 90 nm and 160 to 530 nm).

Our results showed that the amount of platelet-derived EVs appears to be a marker for both short- and long-term outcomes and stroke severity. Several studies reported on increased levels of platelet-derived EVs after a transient ischemic attack and AIS compared with healthy controls [16,17,18,19]. Moreover, this alteration seems to persist or even worsen over time, with the detection of higher platelet-derived EV values than controls up to 90 days after the event [16,17,20]. To the best of our knowledge, only one study assessed the prognostic value of platelet-derived EVs. Rosińska et al. revealed a positive association between elevated concentrations of CD61+ EVs and the recurrence of adverse vascular events in the one-year follow-up period of stroke patients [21]. However, Bivard et al. found that the levels of platelet-derived EVs from blood samples after recanalization treatment were significantly associated with favorable long-term outcomes, assessed by mRS at three months [22]. The disagreement with our finding could derive from the different timing of sampling (after the treatment vs. before the treatment) or of the marker considered for the evaluation of the EV subtype (CD41+ vs. CD61+). A further association was found between the amount of platelet-derived EVs and the atherosclerotic thickening of carotid intima-media in AIS [19,21]. The procoagulant function of platelet-derived EVs has long been known. The procoagulant factors’ density (e.g., phosphatidylserine, CD61, CD62P, and factor X) on the surface of platelet-derived EVs is several times greater than that present on the activated platelets, determining an approximately 50- to 100-fold higher specific procoagulant activity [23]. The increase in the levels of platelet-derived EVs could therefore not only represent a detrimental post-event factor but also represent a pathophysiological mechanism underlying the primary ischemic event.

Regarding endothelial EVs, we found a positive association between CD105+ endothelial-derived EV levels at admission and hypertension in AIS patients. The study of Jimenez and colleagues suggests that CD105+ EV release is associated with endothelial apoptosis [24]. Since the relationship between endothelial dysfunction and hypertension is well known, the release of this EV subtype could represent an important prognostic factor and a useful biomarker for stroke prevention in hypertensive patients [25,26].

To the best of our knowledge, this is the first study that highlights the prognostic value of EVs deriving from T-cells in AIS, with higher amounts of T-cell-derived EVs at admission associated with worse early and mid-term outcomes. There is evidence that T lymphocyte-derived EVs promote endothelial dysfunction through the down-regulation of nitric oxide production and increased oxidative stress in endothelial cells [27,28]. T-cell-derived EVs were found to activate monocytes and to induce degranulation in human mast cells, promoting cytokines release and contributing to a proinflammatory state [27]. Furthermore, it has been found that regulatory T cells (Tregs), a particular subpopulation of T lymphocytes also characterized by CD25 +, have a greater adhesive propensity and favor the interaction of platelets with ischemic endothelial cells of the brain, increasing the risk of further damage into the ischemic area [29]. However, a protective role of Treg cells emerged in stroke patients from the study of Santamaría-Cadavid and colleagues [30]. Higher levels of Treg cells during the acute phase of ischemic stroke were independently associated with smaller infarct volume, prevention of stroke-associated neurological deterioration, and reduction in infections during hospitalization, predicting good functional outcome at 3 months [30]. Finally, the increased levels of T-cell-derived EVs may reflect the degree of the immune and inflammatory response and the resulting tissue damage. Thus, further investigations on the pathophysiological mechanisms of T-cell-derived EVs are needed to clarify their role in AIS.

The temporal differences in the appearance of the two EV subtypes’ prognostic roles revealed a specific temporal profile overlapping with the pathophysiological processes of ischemic stroke: an acute phase linked to ischemic occlusion and platelet and vascular alterations, and a later phase characterized by inflammatory damage [31].

Lastly, patients who did not receive recanalization therapy (pharmacological or mechanical) showed higher levels of EVs at Tw than treated patients. According to what was observed in the Hervella et al. study, this finding could reflect a greater neuroinflammatory response [32]. Since higher cytokine levels in the acute phases of AIS were independently associated with a worse early outcome, but also with greater long-term improvement, and an acute reduction in inflammatory biomarkers was associated with a blunted long-term improvement [32], future research needs to focus on the role of neuroinflammatory processes in relation to treatment, tissue damage, and long-term outcome. Furthermore, the modalities and outcomes of the interaction between the different EV subtypes and the target cells during hypoxic damage are yet to be thoroughly investigated in order to clarify their role and to identify possible new therapeutic approaches. As a matter of fact, EVs contain a wide range of bioactive molecules that can mediate neuroprotection through several mechanisms of action (e.g., antiapoptotic and antioxidative signaling pathways) [1,33].

Our study has some limitations. The first limitation concerns the numerically limited cohort of patients and, in particular, the unequal representation of stroke subtypes. However, having recruited the patients within 4.5 h from the onset of the event, the assessment is very precise and homogeneous in terms of evaluation times and related stroke pathophysiological stage. Future studies should compare the EV profiles and their prognostic role in ischemic versus hemorrhagic stroke at different stages. Second, we cannot exclude that the analytes measured by Nanosight analysis include lipoprotein complexes (very low-density lipoproteins and chylomicrons), such as small cell debris and protein aggregates. However, the flow cytometry analysis can detect only membrane structures that are intact and maintain esterase activity and therefore the detected structures are most likely EVs.

In addition, further investigations will be conducted to clarify the causal relationship between the change in levels of different EV subtypes and the etiopathogenesis and aggravation of ischemic stroke.

## 5. Conclusions

The present study suggests that specific EV subtypes are associated with stroke severity and both short- and long-term outcomes. We found that the platelet-derived EVs at admission were positively associated with the severity of IS at the onset as well as with the mid-term outcome. T-cell-derived EVs at admission were positively related to both early and mid-term IS outcomes and T-cell-derived EVs at T1 were positively related to mid-term IS outcome.

The evidence of our study could contribute to risk stratification in patients with ischemic stroke, to determining the best medical therapy, and to post-recanalization treatment monitoring. The analysis of circulating EVs could therefore represent a useful tool both in emergency and in follow-up evaluation.

## Figures and Tables

**Figure 1 biology-11-01489-f001:**
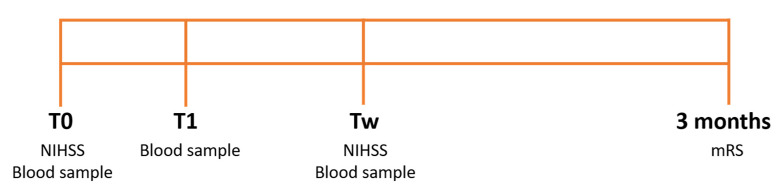
Study protocol. Stroke severity was assessed through the National Institutes of Health Stroke Scale (NIHSS) at admission (T0) and after one week (Tw) and through the Modified Rankin Scale (mRS) after 3 months. Blood samples were collected at T0, after the reperfusion therapy, or 2 h after the first blood sample (T1) and at Tw.

**Figure 2 biology-11-01489-f002:**
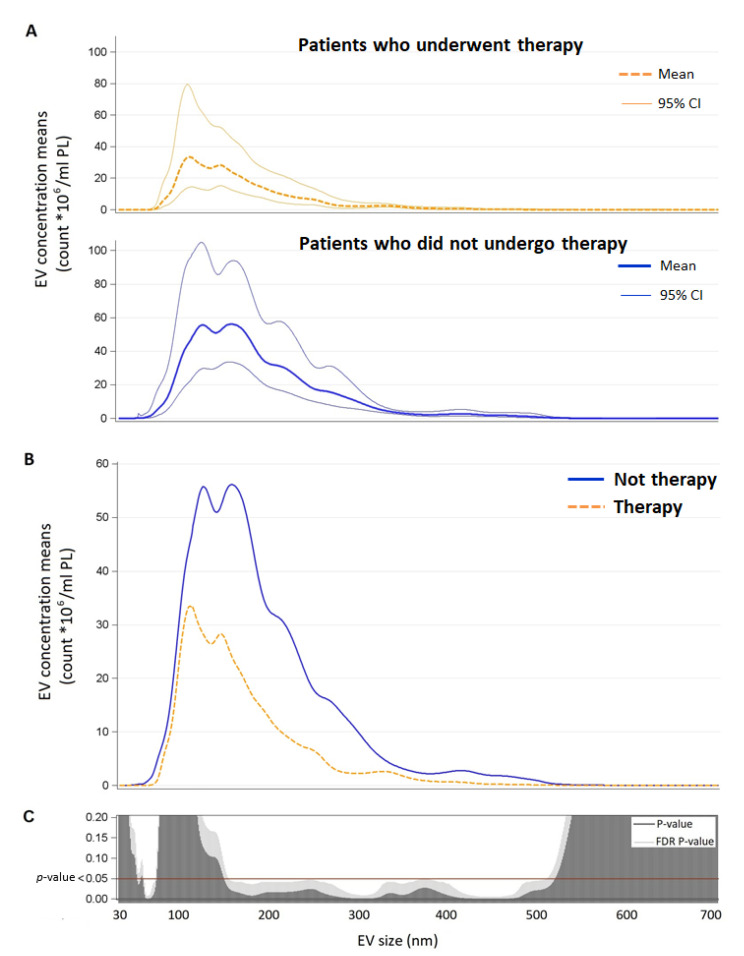
Panel A: EV concentration means and confidence intervals for each EV size (nm) in patients who underwent therapy and patients who did not undergo therapy. Panel B: direct comparison between EV concentration means for each size (nm) by therapy (*n* = 18). Panel C: *p*-value and False Discovery Rate *p*-value for each size are reported. Abbreviations: EV, extracellular vesicles.

**Table 1 biology-11-01489-t001:** Demographic and clinical characteristics of the studied population (*n*= 47).

Characteristics	Value
Age, years	74.1 ± 14.1
Gender *(males)*	26 (55.3%)
BMI, kg/m^2^	24.8 ± 3.8
Previous stroke or TIA	12 (25.5%)
Hypertension	31 (66.0%)
Diabetes	7 (14.9%)
History of atrial fibrillation	15 (31.9%)
History of heart failure	6 (12.8%)
Smoking status	
*Never smoker*	33 (70.2%)
*Current smoker*	10 (21.3%)
*Missing*	4 (8.5%)
Blood pressure, mmHg	
*Systolic*	163 ± 26
*Diastolic*	89 ± 17
Glucose, mg/dL	120.3 ± 32.1
Hb, g/dL	13.9 ± 1.7
Ht, %	40.4 ± 4.3
Plt *10^3^, N/uL	225 [193; 304]
AST, U/L	22.4 ± 7.9
ALT, U/L	18.5 ± 7.5
PT	1.1 ± 0.4
aPTT	0.9 ± 0.1
Albumin, g/dL	4.0 ± 0.4
Creatinine, mg/dL	1.0 ± 0.35
C-reactive protein, mg/dL	0.38 [0.16; 0.75]
Total cholesterol, mg/dL	185.5 ± 34.2
NIHSS on admission	8 [5; 14]
*<14*	34 (72.3%)
*≥14*	13 (27.7%)
Therapy	
*IV rtPA*	18 (38.3%)
*Thrombectomy*	4 (8.5%)
*IV rtPA+ thrombectomy*	6 (12.8%)
*No therapy*	19 (40.4%)
TOAST Classification	
*Cardioembolic stroke*	23 (48.9%)
*Atherothrombotic stroke*	6 (12.8%)
*Lacunar stroke*	2 (4.3%)
*Undetermined etiology*	16 (34.0%)
Hemispheric stroke	
*Right*	22 (46.8%)
*Left*	23 (48.9%)
*Bilateral*	2 (4.3%)
Vascular territory	
*Anterior*	40 (85.1%)
*Posterior*	7 (14.9%)
NIHSS after one week	3 [2; 7]
*<14*	31 (66.0%)
*≥14*	6 (12.8%)
*Missing*	10 (21.2%)
Modified Rankin scale at 3 months	2.8 ± 2.5
*0–2*	20 (42.5%)
*3–5*	10 (21.3%)
*6*	11 (23.4%)
*Missing*	6 (12.8%)

Values are expressed as mean ± standard deviation or as median (Q1, Q3). Abbreviations: BMI, body mass index; TIA, transient ischemic attack; Hb, hemoglobin; Ht, hematocrit; AST, aspartate aminotransferase; ALT, alanine aminotransferase; PT, prothrombin time; aPTT, activated partial thromboplastin time; NIHSS, National Institutes of Health Stroke Scale; TOAST, Trial of Org 10,172 in Acute Stroke Treatment (classification of stroke subtypes based on etiology as the main criterion); mRS, Modified Rankin Scale.

**Table 2 biology-11-01489-t002:** Relationships between National Institute of Health Stroke Scale (NIHSS) at the onset of the acute event and count of EV and EV subtype at T0.

		T0		
**EV Count**	**∆%***	**95% CI**		***p*-Value**
Total EVs	−0.001	−0.006	0.005	0.8317
**EV Subtype**	**∆%****	**95% CI**		***p*-Value**
CD14+ (macrophages/monocytes)	0.044	−0.111	0.199	0.5802
CD61+ (platelets)	**0.177**	**0.065**	**0.289**	**0.0019**
CD105+ (endothelium)	0.497	−1.129	2.150	0.5515
CD25+ (T-cells)	0.255	−1.876	2.432	0.8162
CD62E+ (activated endothelial cells)	−0.567	−1.665	0.544	0.3161

Δ%* = (exp (β ∗ 10^6^) −1) ∗ 100, percentage increase in NIHSS for each 10^6^ increase in EV total count. Δ%** = (exp (β ∗ 10^3^) −1) ∗ 100, percentage increase in NIHSS for each 10^3^ increase in EV subtype count. Abbreviations: EV, extracellular vesicles.

**Table 3 biology-11-01489-t003:** Relationships between National Institute of Health Stroke Scale (NIHSS) one week after the acute event and count of EV and EV subtype at T0, T1, and Tw.

	T0				T1				Tw			
**EV count**	∆%*	**95% CI**		***p*-Value**	**∆%***	**95% CI**		***p*-Value**	**∆%***	**95% CI**		***p*-Value**
Total EVs	0.005	−0.002	0.013	0.1327	0.003	−0.019	0.025	0.7847	0.016	−0.008	0.040	0.1958
**EV subtype**	**∆%****	**95% CI**		***p*-Value**	**∆%****	**95% CI**		***p*-Value**	**∆%****	**95% CI**		***p*-Value**
CD14+ (macrophages/monocytes)	−0.02	−0.21	0.18	0.8763	0.91	−0.44	2.28	0.1857	2.71	−1.3	6.88	0.1877
CD61+ (platelets)	−0.08	−0.39	0.24	0.6363	−0.20	−0.48	0.07	0.1488	−0.81	−3.31	1.74	0.5287
CD105+ (endothelium)	1.86	−0.84	4.63	0.1785	−0.11	−4.09	4.04	0.9571	1.03	−4.38	6.73	0.7161
CD25 + (T-cells)	**3.93**	**0.99**	**6.97**	**0.0086**	−0.79	−2.00	0.44	0.2071	2.4	−3.19	8.31	0.4081
CD62E+ (activated endothelial cells)	0.90	−0.98	2.81	0.3523	2.32	−0.91	5.65	0.1610	4.75	−19.92	37.01	0.7351

Δ%* = (exp (β ∗ 10^6^) −1) ∗ 100, percentage increase in NIHSS for each 10^6^ increase in EV total count. Δ%** = (exp (β ∗ 10^3^) −1) ∗ 100, percentage increase in NIHSS for each 10^3^ increase in EV subtype count. Abbreviations: EV, extracellular vesicles.

**Table 4 biology-11-01489-t004:** Relationships between the modified Rankin Scale (mRS) evaluated 3 months after the acute event and count of EV and EV subtype at T0, T1, and Tw.

	T0				T1				Tw			
**EV count**	**∆%***	**95% CI**		** *p* ** **-value**	**∆%***	**95% CI**		** *p* ** **-value**	**∆%***	**95% CI**		** *p* ** **-value**
Total EVs	0.006	−0.001	0.013	0.0717	−0.017	−0.046	0.013	0.2672	0.016	−0.008	0.040	0.1958
**EV subtype**	**∆%****	**95% CI**		** *p* ** **-value**	**∆%****	**95% CI**		** *p* ** **-value**	**∆%****	**95% CI**		** *p* ** **-value**
CD14+ (macrophages/monocytes)	0.05	−0.10	0.19	0.5517	0.60	−0.33	1.53	0.2073	2.71	−1.3	6.88	0.1877
CD61+ (platelets)	**0.30**	**0.06**	**0.55**	**0.0158**	0.18	−0.03	0.39	0.0958	−0.81	−3.31	1.74	0.5287
CD105+ (endothelium)	1.66	−0.79	4.17	0.1861	2.78	−1.21	6.92	0.1745	1.03	−4.38	6.73	0.7161
CD25 + (T-cells)	**3.94**	**0.96**	**7.01**	**0.0093**	**0.91**	**0.01**	**1.81**	**0.0468**	2.4	−3.19	8.31	0.4081
CD62E+ (activated endothelial cells)	1.79	−0.43	4.06	0.1155	1.92	−1.01	4.94	0.2015	4.75	−19.92	37.01	0.7351

Δ%* = (exp (β ∗ 10^6^) −1) ∗ 100, percentage increase in mRS for each 10^6^ increase in EV total count. Δ%** = (exp (β ∗ 10^3^) −1) ∗ 100, percentage increase in mRS for each 10^3^ increase in EV subtype count. Abbreviations: EV, extracellular vesicles.

## Data Availability

The data presented in this study are available on request from the corresponding author.

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
