# Peer review of "The Role of Extracellular Vesicles in Ischemic Stroke Severity"

_biology, 2022, doi:10.3390/biology11101489_

Round 1

Reviewer 1 Report

1. The article must be formatted in accordance with the requirements of the journal. Simple Summary. abstract. 1.Introduction. 2. Materials and Methods etc

2. Abstract needs to be improved. Emphasis should be placed on your own results.

3. The conclusion should clearly reflect the results obtained. In its present form, the conclusion presents general considerations and perspectives.

4. The quality of Figure 1 needs to be improved. Preferably done in color.

5. It would be great to draw the Experimental protocol in the form of a scheme.

6. The various molecular mechanisms of action of endovesicles need to be discussed. For example https://www.ijbs.com/v18p5345.htm

Author Response

Comments and Suggestions for Authors

  1. The article must be formatted in accordance with the requirements of the journal. Simple Summary. abstract. 1.Introduction. 2. Materials and Methods etc

We thank the Reviewer for this observation. We modified the article according to the requirements of the journal.

  1. Abstract needs to be improved. Emphasis should be placed on your own results.

We thank the Reviewer for the comment. We modified the abstract in order to highlight our results.

  1. The conclusion should clearly reflect the results obtained. In its present form, the conclusion presents general considerations and perspectives.

We thank the Reviewer for his/her observation. We modified the conclusion presenting the results obtained from our study.  

  1. The quality of Figure 1 needs to be improved. Preferably done in color.

We modified the Figure 1 according to the Reviewer’s suggestion.

  1. It would be great to draw the Experimental protocol in the form of a scheme.

We thank the Reviewer for this suggestion. We included a scheme of the experimental protocol with the different timepoint.

  1. The various molecular mechanisms of action of endovesicles need to be discussed. For example https://www.ijbs.com/v18p5345.htm

We thank the Reviewer for his/her observation. Given the complexity of the interactions between different EV subtypes and target cells, it is difficult to establish the specific function of the single biomarker in human studies. However, today more and more preclinical and clinical studies are investigating the mechanism of action and outlining the important diagnostic, prognostic and therapeutic role of EVs in various pathologies and in particular in the cardiovascular and neurological fields. We added a comment and the related references based on the Reviewer’s suggestion in the discussion (lines 327-332).

Reviewer 2 Report

In the present manuscript, the authors have described the correlation between the Extracellular Vesicles (EVs) extracted from the blood, with the severity of the ischemic stroke  (IS) using 47 different patients. The authors also suggested that the EVs may act as the marker of the IS severity. Overall the study is good and vital for the readers in general. I would recommend it for publication after making minor manuscript changes.

1- Please change IS to ischemic stroke in the abstract section as it will confuse the readers. 

2- It would be good for the manuscript if the authors could describe the potency of the IS based on the Hypertension and Smoking status, and calculate EVs concentration for specific markers in these patients exclusively, in addition to the normalized data. This data may give additional information.

3-  Please change PL to plasma in the main text for EVs counts.

Round 2

Reviewer 1 Report

My comments have been taken into account. The article can be accepted for publication in the submitted form.